# Anatomical Variations of the Recurrent Laryngeal Nerve and Implications for Injury Prevention during Surgical Procedures of the Neck

**DOI:** 10.3390/diagnostics10090670

**Published:** 2020-09-04

**Authors:** Alison M. Thomas, Daniel K. Fahim, Jickssa M. Gemechu

**Affiliations:** 1Department of Neurosurgery, Oakland University William Beaumont School of Medicine, Rochester, MI 48309, USA; alison.thomas@beaumont.org (A.M.T.); Daniel.fahim@beaumont.org (D.K.F.); 2Department of Foundational Medical Studies, Oakland University William Beaumont School of Medicine, Rochester, MI 48309, USA; 3Michigan Head & Spine Institute, Southfield, MI 48034, USA

**Keywords:** recurrent laryngeal nerve, inferior thyroid artery, anatomical variations, ACDF, thyroidectomy

## Abstract

Accurate knowledge of anatomical variations of the recurrent laryngeal nerve (RLN) provides information to prevent inadvertent intraoperative injury and ultimately guide best clinical and surgical practices. The present study aims to assess the potential anatomical variability of RLN pertaining to its course, branching pattern, and relationship to the inferior thyroid artery, which makes it vulnerable during surgical procedures of the neck. Fifty-five formalin-fixed cadavers were carefully dissected and examined, with the course of the RLN carefully evaluated and documented bilaterally. Our findings indicate that extra-laryngeal branches coming off the RLN on both the right and left side innervate the esophagus, trachea, and mainly intrinsic laryngeal muscles. On the right side, 89.1% of the cadavers demonstrated 2–5 extra-laryngeal branches. On the left, 74.6% of the cadavers demonstrated 2–3 extra-laryngeal branches. In relation to the inferior thyroid artery (ITA), 67.9% of right RLNs were located anteriorly, while 32.1% were located posteriorly. On the other hand, 32.1% of left RLNs were anterior to the ITA, while 67.9% were related posteriorly. On both sides, 3–5% of RLN crossed in between the branches of the ITA. Anatomical consideration of the variations in the course, branching pattern, and relationship of the RLNs is essential to minimize complications associated with surgical procedures of the neck, especially thyroidectomy and anterior cervical discectomy and fusion (ACDF) surgery. The information gained in this study emphasizes the need to preferentially utilize left-sided approaches for ACDF surgery whenever possible.

## 1. Introduction

The recurrent laryngeal nerve (RLN) is a branch of the vagus nerve (CN X). It carries sensory, motor, and parasympathetic fibers to the laryngeal structures [1]. It is the main motor nerve of all intrinsic laryngeal muscles, except the cricothyroid, which receives its innervation via the external laryngeal nerve [1]. The RLN has a different course on the left and right side of the body, where it loops around the arch of the aorta below the ligamentum arteriosum on the left side, and around the subclavian artery at the base of the neck on the right side [1]. On both sides, after looping deep to the relevant vessels, the RLN follows the tracheoesophageal groove to enter into the larynx to the inferior pharyngeal constrictor muscle [1,2].

It has been reported that anatomical variations of the RLN are commonly more rostral on the right than the left side, which is mainly explained due to the difference in their embryonic origin [1]. The left RLN has a longer course in the thoracic region compared to the right RLN, which makes the former vulnerable to injury due to trauma and masses associated with oncologic diseases [3,4]. Although the right RLN travels within the tracheoesophageal groove, its course is more anterior and lateral than the left RLN, which causes many surgeons and researchers to debate the merits of left or right-sided approaches when performing anterior cervical discectomy and fusion (ACDF) procedures [3,5,6].

The safety of surgical procedures of the neck may be affected by potential anatomical variations of the RLN. In this regard, several variations have been reported in previous studies that make the nerve more vulnerable to damage [1,7]. Bilateral RLN variations in the same patient are also possible, which would increase the vulnerability of the nerve damage during surgical procedures of the neck [3]. Proper identification and preservation of the RLN and consideration of the potential variations may prevent or minimize the damage to the nerve.

Damage to the RLN has been reported as one of the most common iatrogenic complications associated with various surgeries of the neck [7]. RLN damage can be temporary or permanent, depending on the severity of the nerve injury and whether the injury is unilateral or bilateral. RLN damage results in hypophonation, and in extreme cases, dyspnea by paralyzing the muscles of the larynx [7]. Unilateral paralysis to the RLN may be clinically silent and present as hypophonia, dysphonia, or dysphagia and aspiration [8]. The fact that unilateral damage can be clinically silent results in RLN palsy being underdiagnosed and hence under-reported. This has the potential to render patients who undergo multiple procedures of the neck more vulnerable to bilateral damage. Bilateral damage to the RLN typically presents as dyspnea with inspiratory stridor, signifying a narrowed airway [2].

Lack of knowledge regarding the anatomical variations of RLN and its course and branching pattern carries potential risks, compromising the safety of surgical procedures in the neck. This cadaver-based study provides substantial information to close the knowledge gap and elucidate the relationship between the RLN and other anatomical structures in order to improve the safety of neck surgeries. Although previous studies have reported variations regarding the RLN based on a smaller number of cadavers, we endeavored to dissect a large number of cadavers to comprehensively investigate the variations of branching patterns, along with the potential risk of variations in the nerve’s anatomical relationship with the inferior thyroid artery (ITA) [3,7,8,9,10].

## 2. Materials and Methods

The study was performed on 55 formalin-fixed cadavers at Oakland University William Beaumont School of Medicine from 2017–2019, of which 28 were male, 27 were female, and all were Caucasian in ethnicity. Anterior cervical deep dissection was performed in order to locate the RLN and surrounding landmarks on each cadaver. Cadavers that demonstrated anatomical variations were identified, and after detailed clearing, photographs were taken for data analysis. Cadavers without anatomical variations were also photographed for comparison as controls. The data was documented based on whether the left or right nerve was being observed, and also based on the number of branches that were identified on each RLN. We also took into account the relationship of the RLN with the ITA, and whether the nerve was anterior, posterior, or in between the branches of the artery. The data was analyzed quantitatively with the McNemar’s test and Fisher’s exact test, and statistical significance was determined based on a *p* value less than 0.05.

## 3. Results

The findings indicated that there were branches coming off the RLN on both the right and left sides, innervating both the esophagus and trachea in some instances. On the right side, 89.1% demonstrated anywhere between 2–5 extra-laryngeal branches, and 74.6% demonstrated branching on the left side (Table 1, Figure 1). Using the Fisher’s exact test and subsequent data analysis, it was determined that there was a statistically significant difference in the branching pattern between the two sides. Of note, 16.4% of the cadavers demonstrated concomitant bilateral bifurcations in this study, while 16.4% of the cadavers demonstrated concomitant bilateral trifurcations.

Table 1 demonstrates the frequency distribution of branching patterns of the recurrent laryngeal nerve (RLN) on the right and left side. The right RLN branching pattern is documented to be anywhere between 2–5 branches (89.1%), while the left RLN is documented to be 2–5 branches (74.6%). The values were statistically significant, as determined by *p* < 0.05 (McNemar’s test and Fisher’s exact test, *p* = 0.0348).

The relationship of the RLN with the ITA was also examined. The ITA is a common landmark that surgeons use to locate the RLN, especially during thyroidectomy. The RLN was discovered to have a varying relationship with the ITA, and can be seen anterior or posterior to it, and even sometimes in between the branches of the ITA. In relation to the ITA, 67.9% of right RLNs were related anteriorly, while 32.1% were related posteriorly. The opposite values were true for the left side, with 67.9% of the RLNs related posteriorly and 32.1% related anteriorly. On the right side, 3.6% of nerves crossed in between branches of the ITA, while 5.4% of left RLNs were found crossing between branches of the ITA (Table 2, Figure 2).

Table 2 demonstrates the relationship of the recurrent laryngeal nerve (RLN) to the inferior thyroid artery (ITA) on the right and left side. The RLN was observed as either related anteriorly, posteriorly, or in between the ITA. For the right RLN, 67.9% were anterior to the ITA, and 32.1% were posterior to it. The left RLN demonstrated the exact opposite results with 32.1% anterior, and 67.9% posterior to the ITA. The values demonstrated a statistically significant difference, as determined by *p* < 0.05 (McNemar’s test and Fisher’s exact test, *p* = 0.0004).

## 4. Discussion

Injury of the RLN is a well-known and troublesome complication associated with surgical procedures of the neck [9]. Awareness of anatomical variation in the branching patterns of the RLN contributes to the avoidance of this complication. Consideration of these variations is essential to minimize complications associated with surgical procedures of the neck—especially in ACDF and thyroidectomies. Significant postoperative complications, such as airway obstruction and narrowing, can be avoided if surgeons and their assistants in the operating room are aware of these variant structures. Lack of awareness of the anatomical variations in the branching pattern of the RLN makes it vulnerable to damage by stretching, compression, retraction, or accidental sharp division during the aforementioned surgical procedures.

Although a number of studies debate whether the left or right-sided approach to surgery is safer, there is no generally accepted consensus in the surgical community [5,6,8]. To answer this question, our study focused on observed side variations and found that the left RLN branched less often than the right RLN, and also took a more predictable course to the laryngeal structures, rather than an oblique, anterolateral approach. Chen and his colleagues found that overstretching of the RLN is less likely to occur on the left side due to the fact that it is better protected within the tracheoesophageal groove [11]. The right-sided approach is typically taken during an ACDF procedure due to surgeon handedness [11]. The other reason why neurosurgeons, in particular, favor a right-sided approach is due to the awareness of the risk of causing compression of the carotid artery during retraction or the inadvertent dislodging of atherosclerotic plaque during dissection, resulting in a stroke. A right-sided stroke is far better tolerated than a left-sided stroke due to language dominance in the overwhelming majority of patients.

In contrast to the thyroidectomy, in ACDF, the RLN is not routinely monitored or exposed as standard practice, potentially making it more prone to damage from indirect intraoperative injury by retraction or stretch injury while separating fascial layers or during retraction [12]. Of course, the counter-argument is that avoiding exposure of the nerve makes it less vulnerable to injury as no dissection is being carried out immediately around it. Direct surgical trauma to the RLN is rare, which is why injury is typically related to overstretching or excessive pressure from the endotracheal tube [13].

Our findings indicate that there is a significant amount of variation in the course and branching pattern of the RLN, indicating that 89% of right RLNs and 74.6% of left RLNs demonstrated 2–5 extra-laryngeal branches. These findings are in agreement with previous studies that have reported significant variability in RLN branching patterns [14,15,16,17]. This data suggests that the risk of iatrogenic injury is greater with right-sided approaches than left-sided approaches during ACDF procedures. The right RLN has more variability in its branching pattern, relationship with the ITA, and anterolateral position in comparison with the left [16,17]. Because of the fact that extra-laryngeal branching patterns are so common, misidentification of these branches can potentially lead to iatrogenic injury [16]. It is argued that branched nerves are even more vulnerable than those without branches because of their smaller caliber and fragility, ultimately making them prone to damage, even with normal manipulation [12]. Similar to our study, previous findings reported that 50% to 60% of patients often have small branches of the RLN that innervate the trachea, esophagus, or inferior constrictor muscles, and misidentification of these branches can cause a myriad of postoperative symptoms that include dysphagia, dyspnea, and dysphonia [16]. The greater number of branches on the right suggests that the risk of iatrogenic injury is greater with a right-sided approach than with a left-sided approach during ACDF procedures.

In addition to the greater number of branches, the relationship of the right RLN to the ITA may potentially make it more vulnerable to injury as well [16,17]. Campos and Henriques described a typical relationship between the RLN and ITA, where the RLN passed posterior to the nerve [17]. Our findings discredit the idea that the RLN consistently passes posterior to the ITA. Therefore, this idea of the RLN consistently passing posterior to the ITA should be a point of consideration, because it can give surgeons a false sense of security when dissecting anterior to ITA, ultimately leading to an increased risk of damage [13]. Our study, together with findings by other studies, provides evidence demonstrating that the right RLN more often passes anterior to the ITA approximately two-thirds of the time, while the inverse is true on the left [18]. Therefore, left-sided approaches to ACDF procedures may be safer as the ITA can more often be identified before the branches of the RLN are encountered.

In a meta-analysis study regarding anatomic variations of the RLN, a significant difference of 73.3% and 39.2% was noted in the prevalence of extra-laryngeal branch patterns between cadaveric studies and intraoperative studies, respectively [17]. The authors suggested that observed branching of the RLN was underestimated in the operating room, and this was attributed to the difficulty of viewing branches intraoperatively due to localized inflammation and edema that can be encountered in anterior neck dissections [18]. The true prevalence of extra-laryngeal branching was determined to be better reflected in cadaveric studies, prompting surgeons to attempt to expose the RLN and any branches in their entirety, unless it put the patient at risk for a more invasive procedure [17].

Vocal cord paralysis is one of the most significant morbidities after ACDF procedures, with incidence up to 24.2% immediately postoperatively [19]. Identification and localization of the RLN and intraoperative neuromonitoring of clinically relevant anatomical variations must be encouraged, along with the use of loupe magnification to identify the RLN and its landmarks [20]. Henry and others described how pre-operative ultrasound was used to identify structures and anatomical variations successfully [17]. This method successfully identified variants, such as nonrecurrent laryngeal nerves, 98% of the time, making it a reasonable method to decrease the risk of iatrogenic injury to the RLN [17]. Of course, this practice would be time-consuming and potentially cost prohibitive if undertaken for each patient prior to ACDF surgery. We recommend the use of ultrasound evaluation or intraoperative monitoring if a patient has had previous neck surgery. Alternatively, the patient may be subjected to direct laryngoscopy pre-operatively to evaluate for unilateral silent vocal cord paralysis. This population of patients may have clinically silent unilateral RLN injuries and are at higher risk for a potentially devastating injury if the contralateral side is injured in a subsequent procedure of the neck.

In general, the information gained from our study emphasizes the need for special considerations during ACDF and thyroidectomies, including side preferences, in order to preserve the extra-laryngeal branches of the RLN. Parameters that could potentially affect the incidence of branching patterns that must be evaluated further include variations in surgical exposure techniques, retraction practices, the use of surgical loops or magnification, and the use of intraoperative neuromonitoring [18]. The majority of previous studies performed assessing the anatomical variations and landmark structures of the RLN described only one aspect of the nerve and considered smaller sample sizes. Our study tried to address this drawback by considering a comprehensive anatomical approach to show the larger perspective of RLN and its relationship with the surrounding structures using a large sample size of cadavers. One of the limitations of this study is that all the assessed cadavers were Caucasian in ethnicity, and this might affect the generalizability of our findings.

In conclusion, this study may have implications for surgical technique and consideration of the side approach for preserving the extra-laryngeal branches of the RLN during surgical procedures of the neck such as ACDF and thyroidectomy procedures. In general, we recommend preferentially utilizing the left-sided approach for ACDF and being mindful of the proximity of the RLN branches to the ITA. Of course, the side of the surgical approach is multifactorial, and the individualized decision must be made by the surgeon.

## Figures and Tables

**Figure 1 diagnostics-10-00670-f001:**
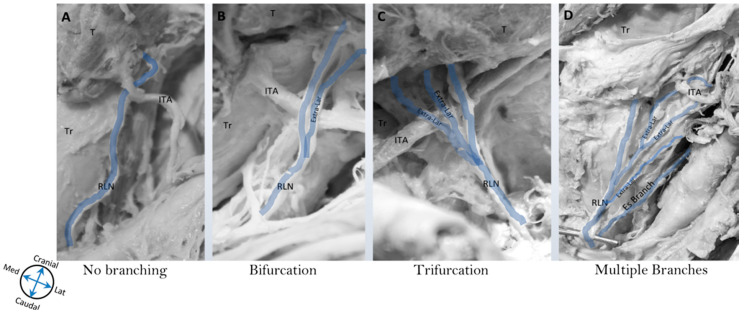
Extra-laryngeal branching pattern of recurrent laryngeal nerve. This figure demonstrates an example of the different branching patterns of the recurrent laryngeal nerve (RLN) documented in this study. (**A**) RLN without extra-laryngeal branching, (**B**) RLN with bifurcation (two branches), (**C**) RLN with trifurcation (three branches), and (**D**) RLN with multiple branches (four branches and above). Abbreviations: Es = esophagus, ITA = inferior thyroid artery, RLN = recurrent laryngeal nerve, T = thyroid, Tr = trachea.

**Figure 2 diagnostics-10-00670-f002:**
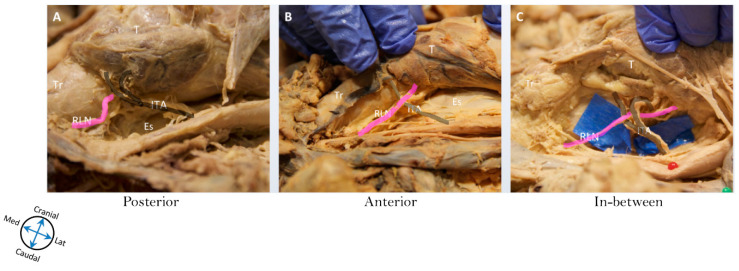
Relationship of recurrent laryngeal nerve to the inferior thyroid artery. This figure demonstrates the various relationships between the recurrent laryngeal nerve (RLN) and inferior thyroid artery (ITA). (**A**) demonstrates RLN related posterior to the ITA, (**B**) demonstrates RLN related anterior to the ITA, and (**C**) demonstrates RLN passing in between the two branches of the ITA. Abbreviations: Es = esophagus, ITA = inferior thyroid artery, RLN = recurrent laryngeal nerve, T = thyroid, Tr = trachea.

**Table 1 diagnostics-10-00670-t001:** Frequency distribution of recurrent laryngeal nerve branching pattern.

	Right Side		Left Side	
	Frequency	%	Frequency	%
Branches		89.09		74.55
Bifurcation (2)	22	40.00	15	27.27
Trifurcation (3)	15	27.27	19	34.55
Multiple (≥4)	12	21.82	7	12.73
No branch (1)	6	10.91	14	25.45

**Table 2 diagnostics-10-00670-t002:** Frequency of relationship of recurrent laryngeal nerve to the inferior thyroid artery.

	Right Side		Left Side	
	Frequency	%	Frequency	%
Anterior	36	67.92	17	32.08
Posterior	17	32.08	36	67.92
In-between	2	3.57	3	5.36

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
