# Peer review of "Anatomical Variations of the Recurrent Laryngeal Nerve and Implications for Injury Prevention during Surgical Procedures of the Neck"

_diagnostics, 2020, doi:10.3390/diagnostics10090670_

Round 1

Reviewer 1 Report

The authors report an interesting cadaver-based study to evaluate variations in the course, branching pattern, and relationship of the recurrent laryngeal nerves (RLNs) to minimize complications associated with surgical procedures of the neck, especially thyroidectomy and anterior cervical discectomy and fusion surgery.

The paper is well documented, scientifically satisfactory, rigorous and well documented.

The bibliography is sufficient but I think it is useful to add the following reference, because I believe that to identification and localization of the RLNs has to be encouraged not only intraoperative neuromonitoring but also the use of microsurgical technique and loupes magnification to enhance visualization of the clinically relevant anatomical variations:

  • D'Orazi V, Panunzi A, Di Lorenzo E, Ortensi A, Cialini M, Anichini S, Ortensi A. “Use of loupes magnification and microsurgical technique in thyroid surgery: ten years experience in a single center”. G Chir 2016 May-Jun;37(3):101-107.

Author Response

Thank you so much for taking the time to review this article and offer recommendations. I included a statement about the use of loupe magnification in line 201, and cited the source you provided. Please see attachment. 

Reviewer 2 Report

Thank you for inviting me to review this interesting paper. It is a solid, well-conducted study with novel findings and clear clinical implications.

Question for the authors to consider: The esophagus is generally ~1/3 skeletal muscle (somatic innervation) proximally, and 2/3 distally smooth muscle (autonomic innervation). The generally assumed fiber composition of the RLN is that it carries primarily somatic motor and somatic sensory fibers. Therefore, in cases where its extralarygneal braches are traveling to the esophagus, do you interpret them as providing only somatic sensory innervation to the esophagus? Or somatic innervation to the skeletal muscle portion of the upper esophagus? Or could it be possible that some autonomic fibers from vagus could be included in the RLN? While I recognize that there’s no way to assess this in a cadaver, I think it has potentially interesting implications. If the branches are only sensory, then damaging them should not directly cause motor deficits of the esophagus such as dysphasia. However, a study you cite [8] seems to indicate that dysphagia may result from RLN paralysis, suggesting that the esophagus receives some of its motor innervation from the RLN. It begs the question as to whether the RLN may sometimes carry parasympathetic fibers as well. I’m not asking the authors to discuss this point necessarily, only that they consider whether a discussion of the fiber composition of the RLN would contribute to the manuscript. I leave it to their discretion.

Minor points:

Line 17: When starting a sentence with a number, please write out the number (e.g., Fifty-five rather than 55).

Lines 19 and 88: “Coming off of” should be “coming off”.

The spelling of extralaryngeal with and without a hyphen varies throughout the manuscript. Please pick one and be consistent (e.g., line 21 vs line 22). A quick Control H in Word could easily solve this.

Line 35: Change “vagus” to “vagus nerve”.

Line 40: Change “beneath” to a more anatomical term, perhaps “deep to”.

Line 48: Change “left one” to “left RLN” or “on the left side”.

Figures 1 and 2: Is there a logic to the order of the listed abbreviations? If not, please list them alphabetically.

Lines 124: For the phrase “statistically significant”, I assume you mean the values between sides were “statistically significantly different”? If so, please change the wording to clarify this point.

Lines 135-136: First sentence of this paragraph needs at least one citation.

Line 141: Change “ignorance” to a less critical-sounding term. How about “lack of awareness”?

Lines 144-145: Needs citations. What are these “number of studies”? Cite them here even if you already mentioned them in the Introduction.

Lines 163-164: Change “in the RLN’s course and branching pattern” to “in the course and branching pattern of the RLN”.

Line 189: “meta-anaylsis” should be “meta-analysis”.

Line 207: I think there’s a missing verb in this sentence. Perhaps “the patient may be subjected to direct laryngoscopy”?

Line 209: Change “opposite side” to “contralateral side”.

Line 211: Change “need of” to “need for”.

Line 214: Changed “looked into” to “investigated” or “evaluated”.

Line 217: Change “RLN’s anatomical variations and landmark structures” to “anatomical variations and landmark structures of the RLN”.

Author Response

Thank you so much for taking the time to review our manuscript and make revisions. I really appreciate the point you brought to our attention about the fiber composition of the esophagus and how the RLN would innervate it. We feel that this information is important to consider as we continue to grow upon this project, and we will address it in more detail in an upcoming manuscript after collecting more data.

I have made all of the minor revisions that you listed, and attached the updated manuscript with tracked changes. 
